# Polyphenols Investigation and In Vitro Antioxidant Capacity of Romanian Wild-Grown *Geranium* spp. (*Geraniaceae*)

**DOI:** 10.3390/plants14203190

**Published:** 2025-10-17

**Authors:** Cornelia Bejenaru, Adina-Elena Segneanu, Andrei Biţă, Ludovic Everard Bejenaru, Marilena-Viorica Hovaneţ, Maria Viorica Ciocîlteu, Adriana Cosmina Tîrnă, Antonia Blendea, George Dan Mogoşanu

**Affiliations:** 1Drug Research Center, Faculty of Pharmacy, University of Medicine and Pharmacy of Craiova, 2 Petru Rareş Street, 200349 Craiova, Romania; cornelia.bejenaru@umfcv.ro (C.B.); andrei.bita@umfcv.ro (A.B.); maria.ciocilteu@umfcv.ro (M.V.C.); tirna.adriana@gmail.com (A.C.T.); antonia.radu@umfcv.ro (A.B.); george.mogosanu@umfcv.ro (G.D.M.); 2Department of Pharmaceutical Botany, Faculty of Pharmacy, University of Medicine and Pharmacy of Craiova, 2 Petru Rareş Street, 200349 Craiova, Romania; 3Institute for Advanced Environmental Research, West University of Timişoara (ICAM–WUT), 4 Oituz Street, 300086 Timişoara, Romania; adina.segneanu@e-uvt.ro; 4Department of Pharmacognosy & Phytotherapy, Faculty of Pharmacy, University of Medicine and Pharmacy of Craiova, 2 Petru Rareş Street, 200349 Craiova, Romania; 5Department of Pharmaceutical Botany and Cell Biology, Faculty of Pharmacy, Carol Davila University of Medicine and Pharmacy, 6 Traian Vuia Street, 020945 Bucharest, Romania; marilena.hovanet@umfcd.ro; 6Department of Instrumental and Analytical Chemistry, Faculty of Pharmacy, University of Medicine and Pharmacy of Craiova, 2 Petru Rareş Street, 200349 Craiova, Romania; 7Doctoral School, University of Medicine and Pharmacy of Craiova, 2 Petru Rareş Street, 200349 Craiova, Romania

**Keywords:** *Geranium* spp., *Geraniaceae*, Romanian flora, phenolic acids, UHPLC/UV–MS analysis, HPTLC fingerprint, antioxidant capacity

## Abstract

*Geranium* spp. are recognized as rich sources of phenolic metabolites, with potential health benefits, yet comparative evaluations remain limited. We assessed four wild-grown *Geranium* spp. (*G. dissectum*—G1, *G. lucidum*—G2, *G. pusillum*—G3, and *G. robertianum*—G4), from southwestern Romanian flora, using complementary antioxidant (DPPH, ABTS, and FRAP) and phytochemical (TPC and TFC) assays. Targeted UHPLC/UV with MS confirmation quantified eight phenolic acids. FRAP provided the strongest discrimination between species, and mirrored TPC, with the highest values in G4 sample. ABTS and DPPH supported the same ranking, and TFC varied only modestly, but differences were narrower and not significant between species. Caffeic acid was highest in G1 sample, and chlorogenic acid was selectively elevated in G3 sample. Gallic and protocatechuic acids were highest in G4 sample, both tracking the FRAP/TPC gradient. Syringic acid and vanillic acid were enriched in weaker antioxidant species. Distinctive signatures included high *p*-coumaric acid in G4 sample and chlorogenic and ferulic acids in G3 sample. Antioxidant potential among *Geranium* spp. is best explained by TPC, particularly hydroxybenzoic acids, with FRAP emerging as the most sensitive discriminator. These findings provide a comparative benchmark for *Geranium* spp. phytochemistry and a framework for future pharmacological studies.

## 1. Introduction

The plants of the *Geraniaceae* family, grouped into six (or seven) genera with over 840 species, are distributed in both hemispheres, mainly in temperate and subtropical zones [1,2,3]. In Europe, the wild-grown *Geraniaceae* belong to only two genera: *Geranium* (39 species) and *Erodium* (34 species) [4]. The two genera are also present in Romania, *Geranium* with 22 species and *Erodium* with three species; eight *Geranium* spp. are frequent, the rest being sporadic or rare [5,6].

Of the more than 400 *Geranium* spp., distributed throughout the Globe, most are present in the Northern Hemisphere, especially in temperate (over 250–300 species) or subtropical areas [1,2,3,5]. The few species from tropical zones grow only in mountainous regions, at high altitudes [1,2]. *Geranium* spp. also grow naturally in northern and southern Africa, Australia, and New Guinea, as well as on various islands in the Pacific and Atlantic, a considerable number of species being cultivated in both Eurasia and North America [2,3].

In the flora of Romania, *Geranium* spp. are herbaceous, annual, biennial, or perennial plants [5,6]. The leaves are petiolate, usually with palmately divided lamina. The flowers of type 5 have obovate or obcordate petals, of white, pink, red, blue, or violet color. The fruit is a syncarp, and at maturity it dehisces into five monospermous mericarps that have a long rostrum curved like a bow [4,5,6].

In fact, the name of the *Geranium* genus comes from the Greek word *geranos*, which means “crane”, referring to the fruit with a long and curved rostrum, resembling a crane’s beak [2,5]. The Romanian vernacular name for the *Geranium* genus is “Ciocul berzei” (Stork’s beak), this being used for most species; however, some are also known by their own names: e.g., *G. robertianum*—“Năpraznic” (Red Robin, herb Robert), and *G. pusillum*—“Buchet” (Bouquet) [5,6].

The phytochemistry of a relatively large number (over 50) of *Geranium* spp. has been studied, some species from temperate zones, such as *G. robertianum*, *G. macrorrhizum*, and *G. thunbergii*, being subjected to more extensive analyses. The chemical composition of *Geranium* spp. includes mainly polyphenolic compounds: ellagitannins, flavonoids, and phenolic acids [7,8,9]. Ellagitannins are represented by geraniin, corilagin, pedunculagin, castalagin, and vescalagin [2,3,7,8,9,10,11,12,13]. The most frequently isolated flavonoids are aglycones (quercetin, kaempferol, and myricetin) and their corresponding glycosides, such as quercitrin, isoquercitrin, spiraeoside, hyperoside, astragalin, rutin, kaempferide, and isokaempferide; proanthocyanidins have also been identified [2,3,7,8,9,10,11,12,13]. Gallic, ellagic, ferulic, caffeic, and chlorogenic acids are the main phenolic acids identified in *Geranium* spp., accompanied by galloyl-, caffeoyl-, feruloyl-, and coumaroyl-quinic derivatives [3,7,8,9,12,13]. The presence of essential oil (linalool, γ-terpinene, germacrene-D, limonene, geraniol, α-terpineol, caryophyllene oxide, and β-selinene) [14,15,16,17,18,19], polysaccharides [20,21], saponins, fatty acids, carotenoids (lutein), lectins, alkaloids, polycarboxylic acids (malic, citric, and tartaric), vitamins (A, B_1_, B_2_, B_3_, C, and E), amino acids, and mineral salts [2,3,7,8,9,13] has also been reported.

Since ancient times, *Geranium* spp. have been used by the populations from the areas where they grow spontaneously, being found in Asian countries, mainly in Chinese and Indian traditional medicine, as well as in the European ethnomedicine and from certain areas of Africa and America [2,3].

In the ethnopharmacology of European countries, *G. robertianum* (herb Robert) is used in several preparations for the treatment of conditions from a quite extensive range of pathologies, such as those of the digestive system (stomatitis, gastritis, ulcer, diarrhea, and digestive and hepatic hemorrhages), respiratory system (sinusitis and influenza), genitourinary tract, and of the cardiovascular system (in arterial hypertension), as well as at the metabolism level (diabetes and hypercholesterolemia), in rheumatism and cancer [2,3,13,22].

While *G. robertianum* is mentioned as a medicinal plant, used for various ailments, in numerous areas of Europe and Asia, but also in North America (USA, Mexico), South America, and Africa (Morocco) [2,3,13,22,23], the other three species studied in this paper are rarely mentioned, *G. dissectum*—in rheumatism (Lebanon), *G. lucidum*—used as a diuretic and astringent (India), and *G. pusillum*—as an analgesic, astringent, in the treatment of wounds (India) [2,3].

*Geranium* spp. have been studied for a wide range of pharmacological properties, as follows: neuroprotective [2,3,13,23,24], antiproliferative [22,25], cytotoxic [2,26,27,28,29], antitumor [2,3,13,30], anti-inflammatory [20,25,31], anti-ulcerative [13,32,33], analgesic and antipyretic [2,3,13,34], antioxidant [2,3,10,11,13,27,31,35,36,37,38], hepatoprotective [21,39], antidiabetic [7,13,40], immunomodulatory [20], antiallergic [12], antibacterial [2,3,13,15,16,17,18,19,27,28,30], antimycotic [27], antiviral [13,35,41,42,43], wound healing [13], antidiarrheal [2,3], anthelmintic [2,3,44], antimalarial [45], anti-leishmanial [2,3], and enzyme-inhibitory [7].

The four *Geranium* spp.—*G. dissectum* L., *G. lucidum* L., *G. pusillum* Burm. f., and *G. robertianum* L.—were selected for study because they are among the most common wild-grown representatives of the genus in the southwestern Romanian flora. In addition, they are readily accessible for comparative sampling, and have limited phytochemical characterization compared to other *Geranium* spp.

The aim of our paper was to investigate, for the first time, the total phenolic content (TPC), total flavonoid content (TFC), phenolic acids profile, and in vitro antioxidant capacity of aerial parts from four wild-grown *Geranium* spp. (*G. dissectum*—G1, *G. lucidum*—G2, *G. pussilum*—G3, and *G. robertianum*—G4) harvested from southwestern Romania flora. 2,2-Diphenyl-1-picrylhydrazyl (DPPH) radical scavenging (half-maximal inhibitory concentration—IC_50_), 2,2′-azino-*bis*(3-ethylbenzothiazoline-6-sulfonic acid) (ABTS) radical scavenging (IC_50_), and ferric-reducing antioxidant power (FRAP; mM Fe^2+^ equivalents) assays were used for the assessment of antioxidant potential of *Geranium* spp. Also, from a therapeutic perspective, the study provides new data for a better comprehension of *Geranium* spp.

This study provides the first comparative assessment of phenolic composition and antioxidant capacity among four wild-grown *Geranium* spp. native to southwestern Romania. While several *Geranium* taxa have been studied globally, data on Romanian wild populations remain scarce. By integrating spectrophotometric assays, high-performance thin-layer chromatography (HPTLC) fingerprinting, and ultra-high-performance liquid chromatography/ultraviolet–mass spectrometry (UHPLC/UV–MS) analysis, this work establishes a regional phytochemical baseline for these species and identifies those with the highest antioxidant potential. These findings are intended to support future pharmacognostic and chemotaxonomic investigations within the Romanian flora.

## 2. Results

The four *Geranium* spp. (G1–G4) were evaluated using antioxidant assays (ABTS, DPPH, and FRAP) and phytochemical analyses (TPC, TFC, and UHPLC/UV–MS). UHPLC/UV–MS analysis identified and quantified eight phenolic acids, with notable findings including high amounts of caffeic acid in G1, chlorogenic and ferulic acids in G3, and elevated gallic, protocatechuic, and *p*-coumaric acids in G4.

### 2.1. ABTS and DPPH IC_50_

The ABTS IC_50_ values of the four *Geranium* spp. ranged from 1.28 ± 0.05 mg/mL in G1 sample to 0.32 ± 0.01 mg/mL in G4 sample. DPPH IC_50_ values ranged from 0.66 ± 0.03 mg/mL (G1) to 0.17 ± 0.01 mg/mL (G4). The two assays produced similar rankings of species; however, pairwise comparisons did not reveal statistically significant differences (*p* > 0.05) (Figure 1a; Table 1).

### 2.2. FRAP Assay

FRAP values ranged from 27.73 ± 0.76 mM Fe^2+^ (G1) to 60.49 ± 1.93 mM Fe^2+^ (G4), with significant differences among all species, establishing the unambiguous order of G4 > G3 > G2 > G1 samples (all pairwise *p* < 0.0001). The stepwise separation is large and consistent across the four samples, and it forms the central biological signal in the dataset: species with higher reducing power stand apart from those with lower reducing power across all comparisons (Figure 1b; Table 1).

### 2.3. TPC and TFC Assay

TPC values ranged from 68.17 ± 1.52 mg GAE/g (G1) to 140.14 ± 3.48 mg GAE/g (G4). TPC assay mirrors FRAP exactly. All six pairwise TPC contrasts were significant (*p* < 0.0001), and the species rank is G4 > G3 > G2 > G1 samples, identical to FRAP. This concordance between TPC and FRAP assays is a defining feature of the dataset (Figure 1c; Table 1).

TFC values showed minor variation across species, ranging from 4.06 ± 0.09 mg QE/g (G2) to 6.72 ± 0.23 mg QE/g (G3). By contrast, TFC assay did not differ among species; all pairwise comparisons were non-significant (*p* > 0.05). The absence of a TFC signal, despite a strong TPC/FRAP pattern, suggests that non-flavonoid phenolics are the primary drivers of reducing capacity in these extracts (Figure 1d; Table 1).

### 2.4. HPTLC Fingerprinting and Effect-Directed DPPH Assay

Each extract (G1–G4 samples) was applied in duplicate on silica gel HPTLC plates alongside reference standards of caffeic acid and chlorogenic acid (left margin). Plates were documented at 254 nm (Figure 2) and 365 nm (Figure 3) and then subjected to an effect-directed DPPH assay; radical scavenging zones appear as bleached (pale/yellow) bands on a purple background (Figure 4).

All four *Geranium* spp. showed rich phenolic fingerprints, with multiple UV-absorbing zones (254 nm) spanning low to high retention factor (R_f_). A compact, low-mobility zone co-migrated with chlorogenic acid and was strongest visible in G2 and G4 samples, and only faint in G3 sample. A mid-mobility band was prominent in G1 sample and moderate in G2 sample. Beyond these standards, G4 and G3 samples displayed a cluster of mid-to-low R_f_ bands absent or weaker in the other species, while G1 sample showed several low and high R_f_ bands matching its caffeic acid profile (Figure 2).

At 365 nm, band color and intensity patterns differentiated the *Geranium* spp. G3 sample exhibited a bright, low-R_f_ fluorescent zone at the chlorogenic acid position and a set of green/blue bands at mid-R_f_. The same band at the chlorogenic acid position was spotted also in G2 and G4 samples. G4 sample showed a distinctive, more intense mid-to-upper R_f_ band cluster, consistent with a higher load of less-polar phenolics. G1 and G2 samples were dominated by lower- and mid-R_f_ zones (including the caffeic acid position in G1 sample), with comparatively less intensity at the upper part of the HPTLC plate. The overall scarcity of bright, flavonoid-like fluorescence bands across lanes matches the absence of between-species differences in TFC from the spectrophotometric assay (Figure 3).

Upon DPPH treatment, all tracks showed distinct bleaching zones, confirming radical scavenging constituents are present in each extract. The number and intensity of bleaching zones were greatest in G4 and G3 samples, distributed across mid and upper R_f_ regions; G1 sample showed strong bleaching between the caffeic acid and chlorogenic acid positions and weaker, fewer zones elsewhere; G2 sample exhibited a simpler pattern, with fewer, less intense active bands. Notably, a pronounced bleached band at the chlorogenic acid position was prominent in G3 sample, while G4 sample displayed multiple strong bleached bands not co-migrating with caffeic or chlorogenic acids, indicating additional radical scavenging constituents in that extract (Figure 4).

### 2.5. Phenolic Acids Profile (UHPLC/UV–MS Analysis)

Targeted quantification of eight phenolic acids revealed two broad motifs: one represents acids that are according to the FRAP/TPC gradient and are abundant in the stronger species, and the other are acids that are enriched in the weaker species and show a negative correlation (Figure 5 and Figure 6; Table 2).

Firstly, regarding the caffeic acid, all the pairwise comparisons were significant (*p* < 0.0001), with the rank of G1 > G2 > G4 > G3 samples. Caffeic acid is therefore most abundant in *Geranium* spp., with lower FRAP/TPC and least abundant in the top performers. Chlorogenic acid has a distinctive signature of G3 sample, exceeding all other species (*p* < 0.0001), while G1, G2, and G4 samples did not differ from each other (*p* > 0.05). *p*-Coumaric acid is among the most striking separations in the study, with all pairwise comparisons significant (*p* < 0.0001) and a pronounced rank of G4 > G1 > G3 > G2 samples. The extreme elevation in G4 sample marks *p*-coumaric acid as a characteristic feature of the top FRAP/TPC species. Ferulic acid has a nuanced pattern—G3 > G1 > G4 ≈ G2 samples—with G2 vs. G4 samples specifically non-significant (*p* = 0.592), and the other pairwise contrasts reaching significance. For gallic acid, all pairs are significant (*p* < 0.0001), with G4 > G3 > G2 > G1 samples, exactly matching the FRAP and TPC ranks. Also, for protocatechuic acid, all the pairs are significant (*p* < 0.0001), of G4 > G3 > G2 > G1 samples, aligning with the FRAP/TPC gradient. Syringic acid has all significant pairs (mostly *p* < 0.0001), with G1 > G2 > G4 > G3 samples. This phenolic acid peaks in the weaker antioxidant species. For vanillic acid, also all pairs are significant (*p* < 0.0001), with G1 > G2 > G3 > G4 samples, again concentrated in the lower-activity species.

These profiles highlight a benzoic acid-rich pattern (e.g., gallic acid and protocatechuic acid) in the strongest species, and higher syringic acid/vanillic acid in the weaker species (Figure 5 and Figure 6; Table 2).

### 2.6. Relationships Between Phenolic Profiles and Antioxidant Capacity

ABTS and DPPH assays were nearly identical (*p* = 0.002), hence their shared lack of between-species discrimination. Both IC_50_ readouts trended negatively with FRAP and TPC assays (consistent with the inverse meaning of IC_50_), although, with only four species, these trends were just above conventional thresholds (*p* ≈ 0.06–0.11). The central relationship is the near-perfect association of FRAP with TPC (*p* = 0.006), which directly supports the conclusion that total phenolics explain the species hierarchy in reducing power. Among individual phenolic acids, gallic and protocatechuic acids were the most informative: both rose with FRAP/TPC (*p* ≤ 0.040 across the key pairs). Syringic and vanillic acids showed the opposite pattern: both were positively associated with ABTS/DPPH IC_50_ (*p* ≤ 0.034) and negatively with FRAP (vanillic acid, *p* = 0.013), indicating enrichment in species with weaker antioxidant performance. *p*-Coumaric acid aligned moderately with FRAP/TPC but served primarily as a strong species marker for G4 sample; caffeic acid aligned with the weaker species; chlorogenic and ferulic acids behaved as species-specific signatures rather than general activity drivers.

## 3. Discussion

Taken together, the data present a clear, biologically plausible picture of how antioxidant capacity varies among four *Geranium* spp., and which chemical features are most closely associated with those differences.

The findings of our research closely align with previous studies on *Geranium* spp., regarding their TPC, TFC, phenolic acid composition [13,26,46,47,48,49,50,51], and antioxidant potential [7,11,13,31,36,52].

### 3.1. Correlation Between TPC, TFC, and Antioxidant Capacity

Firstly, reducing capacity and total phenolics define the species hierarchy. The most robust signal is the good agreement between FRAP and TPC: both rank the four *Geranium* spp. (G4 > G3 > G2 > G1 samples), and both separate every pair strongly (*p* < 0.0001 for all TPC and FRAP contrasts). Their good association (*p* = 0.006) is exactly what would be expected if the overall abundance of redox-active phenolics were the dominant driver of reducing power under the FRAP paradigm. This agreement gives the study a solid backbone: FRAP/TPC tell the same story from different angles—functional reducing capacity and chemical pool size.

Secondly, ABTS and DPPH are consistent but non-discriminating here. ABTS and DPPH readouts are internally consistent with each other (*p* = 0.002) and directionally consistent with FRAP/TPC (trending negative), yet they did not detect between-species differences at the tested concentration ranges (all pairwise *p* > 0.05). This outcome is not contradictory; it reflects assay physics and dynamic range. FRAP measures electron transfer to a ferric complex in a relatively uniform chemical environment and tends to scale with TPC. ABTS and DPPH involve radical quenching, where kinetic access and specific structural motifs influence the apparent potency; when the actual differences among species fall within a narrow band relative to variance, pairwise tests will not separate them, even if trends vs. FRAP/TPC are in the expected direction. The correct statement is that no significant differences were detected for ABTS/DPPH under these conditions [53,54,55,56].

Lastly, the phenolic acids segregate into two functional groupings with respect to antioxidant capacity, being possible drivers of the FRAP/TPC gradient. Gallic and protocatechuic acids are parallel to the FRAP/TPC hierarchy at the species level (both all pairs significant, *p* < 0.0001) and rise together with reducing power and total phenolics (*p* ≤ 0.040 for key associations). Chemically, these are hydroxybenzoic acids with substitution patterns favorable to electron donation and radical stabilization, which align with their strong mapping to FRAP. Their co-variation (gallic acid–protocatechuic acid, *p* = 0.046) and their exact match to the FRAP/TPC ranks point to these compounds as major contributors to the observed interspecific differences in reducing capacity [54,55]. Syringic and vanillic acids seem to be negative markers of antioxidant strength as they are most abundant in the *Geranium* spp. with lower FRAP/TPC (G1, then G2 sample) and are positively associated with ABTS/DPPH IC_50_ (*p* ≤ 0.034), while being negatively associated with FRAP (vanillic acid, *p* = 0.013). These patterns suggest that syringic acid/vanillic acid enrichment accompanies weaker antioxidant performance in this set of species. Mechanistically, the additional methoxy substitution relative to gallic acid may shift redox potential and alter reaction kinetics with radicals [57].

*p*-Coumaric acid is exceptionally elevated in G4 sample (all pairs *p* < 0.0001), supporting its role as a chemotaxonomic marker for that species and a plausible contributor to its total phenolics. Caffeic acid peaks in G1 sample, consistent with that species’ lower FRAP/TPC. Chlorogenic and ferulic acids are most characteristic of G3 sample and co-vary closely with each other; their associations with FRAP/TPC are modest and species-specific rather than global drivers of antioxidant strength across all four *Geranium* spp.

The lack of significant TFC differences (all pairwise *p* > 0.05) and only modest associations with FRAP or TPC indicates that non-flavonoid phenolics dominate the observed pattern in reducing capacity. This finding is consistent with the strong benzoic acid signal (gallic acid and protocatechuic acid) and the minimal explanatory value of TFC in distinguishing species. It does not exclude an important role for flavonoids in other antioxidant contexts (e.g., cellular assays or metal chelation), but within this extraction/assay framework, flavonoids do not explain the species hierarchy.

These findings suggest that the dominance of non-flavonoid polyphenols in FRAP/TPC assays is mechanistically linked to their structural features. Hydroxybenzoic acids, such as gallic and protocatechuic acids, possess *ortho*-dihydroxyl substitution patterns that favor electron donation and stabilization of semiquinone radicals, thereby enhancing reducing power in electron transfer-based assays such as FRAP. In contrast, syringic and vanillic acids, enriched in the weaker antioxidant species, contain additional methoxy substitutions that shift redox potential and can slow radical-quenching kinetics, leading to a lower contribution to overall reducing capacity. This interpretation is consistent with previous reports demonstrating that benzoic acid derivatives often surpass flavonoids in driving total reducing power when assessed by Folin–Ciocalteu and FRAP assays in plant extracts [8,9,10,11,35,37]. Thus, while flavonoids remain important contributors in other antioxidant contexts, the present results reinforce that non-flavonoid phenolics are the principal determinants of reducing capacity in *Geranium* spp. under the tested extraction and assay conditions.

From a practical standpoint, the combination of high TPC, high FRAP, and elevated gallic acid/protocatechuic acid levels makes G4 sample the most promising candidate as a source of antioxidant phenolics in this set, with G3 sample a close second on the basis of both benzoic acids and its distinctive chlorogenic acid/ferulic acid profile. G2 sample sits in an intermediate position, and G1 sample is characterized by low TPC/FRAP together with high syringic acid/vanillic acid and caffeic acid content—features that align with weaker reducing capacity. For applications that value total reducing power and benzoic acid-rich profiles, G4 sample appears to be the best starting point.

The co-migration of sample bands with the caffeic and chlorogenic acids standards mirrors the UHPLC results: caffeic acid was highest in G1 sample (*p* < 0.0001 across all pairs), and chlorogenic acid was selectively elevated in G3 sample (G3 > all others, *p* < 0.0001; the remaining species did not differ). On the plates, the caffeic acid-aligned zone is most intense in G1 sample, and the chlorogenic acid-aligned zone is most intense in G3 sample, exactly as expected. Beyond those two standards, the richer mid-to-upper R_f_ pattern in G4 sample aligns with its very high *p*-coumaric acid (all pairwise *p* < 0.0001) and its elevated gallic acid/protocatechuic acid levels. Conversely, the lower-to-mid R_f_ emphasis in G1 sample matches its enrichment in caffeic, syringic, and vanillic acids.

The FRAP and TPC cleanly rank *Geranium* spp. (G4 > G3 > G2 > G1 samples, all pairwise *p* < 0.0001) and are tightly associated (FRAP/TPC, *p* = 0.006), while ABTS/DPPH IC_50_ did not differ between species (*p* > 0.05) despite trending opposite to FRAP/TPC. This apparent discrepancy is also clarified by the HPTLC–DPPH plate. Firstly, active zones exist in all extracts, but G4 and G3 samples carry more (and stronger) individual scavengers, as seen by the broader, deeper bleaching across mid/high R_f_. When such complex mixtures are assayed in bulk, antagonistic/synergistic interactions and a narrow dynamic range of IC_50_ can flatten between-group differences, even though the underlying chemistry differs. In contrast, FRAP (electron-transfer in a uniform redox system) scales directly with the total pool of redox-active phenolics—which is higher in G4 and G3 samples—so FRAP resolves the species cleanly. The HPTLC plate therefore provides a visual, compound-resolved complement to the FRAP/TPC relationship: G4 and G3 samples are not only richer in total phenolics, but they also contain multiple discrete radical-quenching constituents.

The HPTLC plates visually confirm identity for caffeic and chlorogenic acids and show that some of the most active bands in G4/G3 sample are not those two standards, dovetailing with the strong roles of gallic, protocatechuic, and *p*-coumaric acids from UHPLC. They also localize the radical scavenging activity into discrete constituents (multiple bands per species), which explains why solution-phase ABTS/DPPH can under-discriminate when activities average out across the mixture. Finally, they provide a chemotaxonomic fingerprint: G4 sample with strong mid/high-R_f_ clusters, G3 sample with a dominant low-R_f_ chlorogenic acid zone plus mid-R_f_ bands, G1 sample with pronounced caffeic acid-aligned activity and fewer additional zones, and G2 sample intermediate—precisely the qualitative order seen for FRAP/TPC.

### 3.2. Study Limitations

This work compares only four *Geranium* spp., sampled once and analyzed with triplicate replicates, which limits statistical power—especially for assays with narrow dynamic ranges—and constrains generalization beyond the materials, organ (aerial parts), and harvest conditions studied. Antioxidant endpoints are in vitro, chemistry-based surrogates (DPPH, ABTS, and FRAP) that do not model bioavailability, metabolism, or cellular context; Folin–Ciocalteu (TPC) is not specific to phenolics and can respond to other reductants. A single extraction protocol and solvent system were used, so results reflect an extractable phenolic fraction rather than total tissue chemistry; alternative solvents/parameters could shift both yields and profiles. Targeted quantification focused on eight phenolic acids, omitting larger contributors such as ellagitannins, proanthocyanidins, and diverse flavonoid glycosides.

## 4. Materials and Methods

### 4.1. Plant Material

The flowering aerial parts of wild-grown *Geranium* spp. were harvested during the spring period (April 2024) from southwest Romania flora (Oltenia Region). The vegetal samples for analysis were deposited in the Herbarium of the Department of Pharmaceutical Botany, Faculty of Pharmacy, University of Medicine and Pharmacy of Craiova. Twenty-four hours before processing for extraction and analysis, the plant material was first air-dried and then deposited in brown paper bags, at room temperature (RT), in a cool and dark area. Endangered or protected herbal species are not included in our research. The four samples were denoted using a systematic notation to represent different *Geranium* spp., vegetal product (*herba*), date/collection site, geographic coordinates, and voucher specimen. This notation facilitated a clear and organized reference to the specific plant species and part analyzed in the experiments (Table 3).

### 4.2. Chemicals and Reagents

The solvents used in this study included ethanol, methanol, ethyl acetate, and acetonitrile (Merck, Darmstadt, Germany). Ultrapure water was produced using a HALIOS 6 laboratory water system (Neptec, Montabaur, Germany). To improve the performance of the mobile phases for HPTLC and UHPLC analysis, formic acid (Merck) was added.

For TPC, TFC, and antioxidant assays, chemicals and reagents from Sigma-Aldrich (Taufkirchen, Germany) were used as follows: Folin–Ciocilteu reagent, sodium carbonate, gallic acid, aluminum chloride (AlCl_3_), quercetin, ferrous sulfate heptahydrate (FeSO_4_·7H_2_O), DPPH, ABTS, potassium persulfate, sodium acetate, acetic acid, 2,4,6-*tris*(2-pyridyl)-*s*-triazine (TPTZ), hydrochloric acid (HCl), ferric chloride (FeCl_3_), and natural product–polyethylene glycol (NP–PEG) reagent.

For the UHPLC analysis, a set of eight phenolic acid standards was used for both calibration and compound identification. Caffeic acid, chlorogenic acid, *p*-coumaric acid, ferulic acid, gallic acid, protocatechuic acid, syringic acid, and vanillic acid were used as analytical standards and supplied by Merck Millipore (Darmstadt, Germany).

Silica gel 60 F_254_ glass plates (20 × 10 cm), obtained from Merck (Darmstadt, Germany), were used for HPTLC analysis.

### 4.3. Extraction Procedure

The extraction of plant material was carried out using an ultrasound-assisted extraction method, with 70% ethanol as the solvent. A sample of 1 g of finely ground plant material was combined with 10 mL of the ethanol solution. This mixture was then treated in a Bandelin Sonorex Digiplus DL 102H ultrasound bath (Bandelin electronic GmbH & Co. KG, Berlin, Germany). The treatment lasted 20 min at a constant temperature of 50 °C. For UHPLC analysis, the sample was filtered through a 0.22 μm membrane filter into appropriate vials to obtain a clear extract suitable for chromatographic injection.

### 4.4. Standards Preparation

Caffeic acid, chlorogenic acid, *p*-coumaric acid, ferulic acid, gallic acid, protocatechuic acid, syringic acid, and vanillic acid were used as standards for the UHPLC analysis. A stock solution of each standard was prepared at 1 mg/mL concentration using methanol. To achieve calibration concentrations ranging from 0.1 μg/mL to 50 μg/mL, serial dilutions were made. For both standards and samples, a volume of 10 μL was injected into the UHPLC system.

### 4.5. Antioxidant Capacity Assays

#### 4.5.1. DPPH Antioxidant Assay

Using a 96-well microplate, the DPPH radical scavenging assay began by adding 50 μL of each sample, which was then serially diluted to achieve a gradient of decreasing concentrations. After this, 150 μL of a 0.2 mM DPPH solution in ethanol was added to each well. The mixtures were then incubated in the dark at RT for 30 min. The absorbance of each well was measured at 517 nm with a FLUOstar Optima microplate reader (BMG Labtech, Ortenberg, Germany). The antioxidant potential was determined by calculating the IC_50_, which corresponds to the concentration required to scavenge 50% of the DPPH• radicals. To ensure accurate results, each sample was tested in triplicate [58].

#### 4.5.2. ABTS Antioxidant Assay

In the ABTS radical scavenging assay, 50 μL of each sample was added to a 96-well microplate. The samples were then serially diluted, similar to the DPPH assay, to obtain a concentration gradient. Next, 150 μL of an ABTS solution was added to each well. ABTS solution was made by combining 7.4 mM ABTS with 2.6 mM potassium persulfate. The absorbance was measured at 620 nm using a FLUOstar Optima microplate reader (BMG Labtech). The IC_50_ value was then calculated from a dose–response curve, which shows the sample concentration needed to inhibit 50% of the ABTS•^+^ radical cations. All samples were analyzed in triplicate [59].

#### 4.5.3. FRAP Antioxidant Assay

The FRAP assay was performed using freshly prepared FRAP reagent, consisting of acetate buffer, 10 mM TPTZ in 40 mM HCl, and 20 mM FeCl_3_ solution. A calibration curve (*y* = 0.0006123*x* + 0.05584, *R*^2^ = 0.9975) was created using Fe^2+^ standards ranging from 31.2 to 1000 μM. For each assay, 10 μL of sample or standard was added to a 96-well microplate, followed by 190 μL of freshly prepared FRAP reagent. The resulting mixtures were incubated at RT for 30 min before the absorbance was measured at 595 nm. The final results were expressed as mM Fe^2+^ equivalents. All measurements were performed in triplicate to ensure accuracy and reproducibility [59].

### 4.6. Total Polyphenols and Flavonoids

#### 4.6.1. TPC Assay

The TPC of plant extracts was determined using the Folin–Ciocalteu method. Briefly, 20 μL of each extract was added to a 96-well microplate, followed by 100 μL of Folin–Ciocalteu reagent. After mixing for 3 min, 80 μL of 4% sodium carbonate solution was added, and the plate was mixed again for uniformity. The reaction mixtures were incubated in the dark for 2 h. Absorbance was measured at 620 nm using a FLUOstar Optima microplate reader (BMG Labtech). To quantify the phenolic compounds, a gallic acid standard curve (*y* = 0.003056*x* − 0.06985, *R*^2^ = 0.9914) was prepared over a concentration range of 31.25 μg/mL to 1 mg/mL, and results were expressed as mg gallic acid equivalents (GAE) per g of dry sample. All measurements were performed in triplicate to ensure accuracy and reproducibility [58].

#### 4.6.2. TFC Assay

The TFC of plant extracts was determined using an AlCl_3_ colorimetric assay. A quercetin standard curve (*y* = 0.008502*x* + 0.05496, *R*^2^ = 0.9887) was prepared with concentrations from 3.125 to 100 μg/mL in 96% ethanol. For each assay, 50 μL of plant extract or a quercetin standard solution was added to a 96-well microplate, followed by 10 μL of 10% AlCl_3_ solution. Subsequently, 150 μL of 96% ethanol and 10 μL of 1 M sodium acetate were added. A blank control was prepared by replacing the sample with 96% ethanol. The plate was thoroughly mixed and incubated in the dark at RT for 40 min. Absorbance was recorded at 410 nm using a FLUOstar Optima microplate reader (BMG Labtech). The results were expressed as mg quercetin equivalents (QE) per g of dry sample, and each sample was tested in triplicate to ensure reproducibility [58,60].

### 4.7. HPTLC Fingerprinting for Antioxidant Capacity

HPTLC fingerprinting was used to evaluate the antioxidant potential (DPPH assay) of the plant extracts, using caffeic acid and chlorogenic acid as reference standards. For sample applications, a Linomat 5 applicator was used to apply 2 μL of each extract and standard onto the HPTLC plates. Chromatographic separation was performed in a twin trough chamber with a mobile phase composed of ethyl acetate, formic acid, and water (90:6:9, *v*/*v*/*v*). The chamber was saturated for 20 min before development to ensure ideal separation, and the plates were developed until the solvent front reached 7 cm. After development, the plates were air-dried at RT. Visualization of the HPTLC plates was initially performed at 254 nm and 366 nm without derivatization. After derivatization, plates were visualized with NP–PEG reagent, at 366 nm, while the DPPH was observed under white light. This approach facilitated the identification of bioactive compounds based on their R_f_ values and characteristic color changes, indicative of their antioxidant properties [61].

### 4.8. UHPLC Analysis of Phenolic Acids

UHPLC analysis was performed using a Waters Acquity Arc system equipped with both a photodiode array (PDA) detector and a QDa mass detector (Waters, Milford, Massachusetts, USA). The compounds were separated using a CORTECS C18 column (4.6 × 50 mm, 2.7 μm particle size) kept at 28 °C. The mobile phase was a mixture of water with 0.01% formic acid (A) and acetonitrile with 0.01% formic acid (B). Gradient elution was applied, beginning with 99% A at a flow rate of 0.8 mL/min for 1 min, followed by a gradual decrease to 70% A over 1–13 min, which was maintained until 13.10 min. For column washing, the mobile phase was adjusted to 20% A from 13.60 to 17.60 min to elute strongly retained compounds. The system was then re-equilibrated by returning to 99% A at 18.10 min and holding it until 21.10 min. To ensure analytical stability, the column was equilibrated for 10 min between injections, and samples were stored at 8 °C. Compound quantification was performed using absorbance detection at two wavelengths: gallic acid (retention time (t_R_) 1.90 min), protocatechuic acid (t_R_ 3.60 min), vanillic acid (t_R_ 6.10 min), and syringic acid (t_R_ 6.60 min) at 265 nm, and chlorogenic acid (t_R_ 5.80 min), caffeic acid (t_R_ 6.30 min), *p*-coumaric acid (t_R_ 7.80 min), and ferulic acid (t_R_ 8.50 min) at 325 nm. Mass confirmation for definitive compound identification was performed in negative ion mode, targeting specific mass-to-charge (*m/z*) ratios: 153 for protocatechuic acid, 163 for *p*-coumaric acid, 167 for vanillic acid, 169 for gallic acid, 179 for caffeic acid, 193 for ferulic acid, 197 for syringic acid, and 353 for chlorogenic acid [59,61].

### 4.9. Statistical Analysis

All analyses were performed in GraphPad Prism version 9.1.0 (GraphPad Software, San Diego, CA, USA). A two-way analysis of variance (ANOVA) model was applied with species (G1–G4 samples) and assay type (ABTS, DPPH, FRAP, TPC, and TFC) as independent variables, allowing assessment of both main effects and interaction effects. *Post hoc* pairwise comparisons among species were performed within each assay using Tukey’s multiple comparisons test (α = 0.05). Data are expressed as mean ± standard deviation (SD). Biological replication was set as *n* = 3 independent extractions per species, each measured in triplicate. Significance levels are indicated in the results with not significant (ns), * *p* < 0.05, ** *p* < 0.01, *** *p* < 0.001, and **** *p* < 0.0001.

Before group comparisons, assumptions were evaluated on model residuals: normality (Shapiro–Wilk) and homogeneity of variance (Brown–Forsythe/Levene). All datasets met the criteria, so we used two-way ANOVA followed by Tukey’s multiple comparisons (*α* = 0.05).

Associations between continuous variables (e.g., FRAP, TPC, TFC, and individual phenolic acids) were assessed with Pearson’s correlation when normal.

## 5. Conclusions

For the first time, we assessed four wild-grown *Geranium* spp. (*G. dissectum*, *G. lucidum*, *G. pusillum*, and *G. robertianum*), from southwestern Romanian flora, using complementary antioxidant (DPPH, ABTS, and FRAP) and phytochemical (TPC and TFC) assays. Reducing capacity (FRAP) and TPC define a shared species hierarchy (G4 > G3 > G2 > G1 samples) and are tightly linked (*p* = 0.006). ABTS/DPPH were internally consistent (*p* = 0.002) but non-discriminating at the tested ranges (all pairwise *p* > 0.05). Within the phenolic acid pool, gallic and protocatechuic acids align with strong antioxidant profiles, while syringic and vanillic acids are enriched in weaker profiles, with additional species-specific signatures for *p*-coumaric acid (G4 sample) and chlorogenic acid/ferulic acid (G3 sample). These results collectively prioritize G4 sample—and secondarily G3 sample—as sources of antioxidant phenolics and provide clear chemical targets for future fractionation or bioactivity-guided work.

## Figures and Tables

**Figure 1 plants-14-03190-f001:**
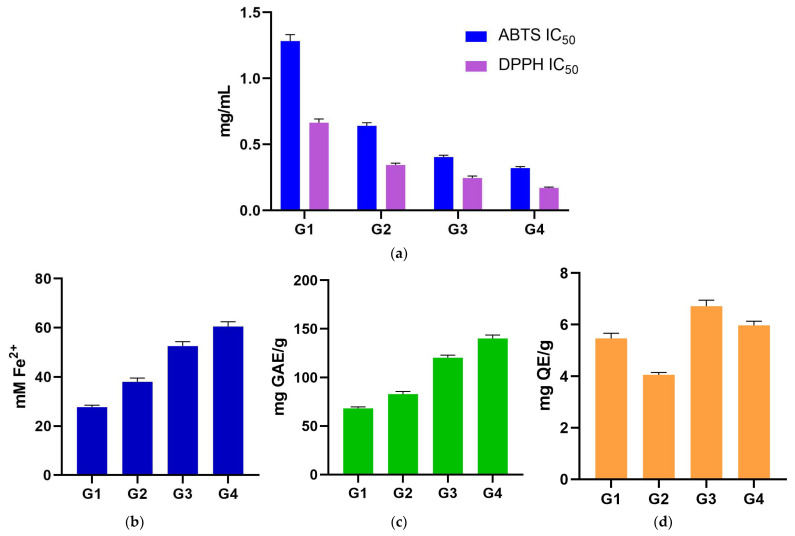
(**a**) ABTS and DPPH assays (grouped IC_50_ plots)—pairwise differences were not detected (*p* > 0.05); (**b**) FRAP antioxidant assay; (**c**) TPC assay, underscoring the matched order with FRAP assay; (**d**) TFC assay stating that all pairwise contrasts were non-significant (*p* > 0.05). ABTS: 2,2′-Azino-*bis*(3-ethylbenzthiazoline sulfonic acid); DPPH: 2,2-Diphenyl-1-picrylhydrazyl; FRAP: ferric-reducing antioxidant power; G1: *G. dissectum*; G2: *G. lucidum*; G3: *G. pussilum*; G4: *G. robertianum*; GAE: gallic acid equivalents; IC_50_: half-maximal inhibitory concentration; QE: quercetin equivalents; TFC: total flavonoid content; TPC: total phenolic content.

**Figure 2 plants-14-03190-f002:**
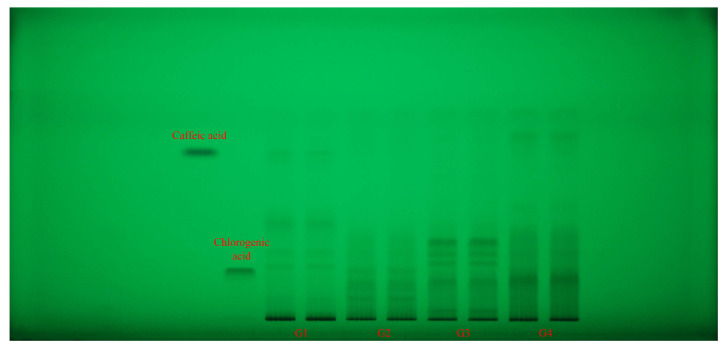
HPTLC fingerprint (254 nm, without derivatization) for *Geranium* (G1–G4) samples and phenolic acid reference bands. HPTLC: high-performance thin-layer chromatography.

**Figure 3 plants-14-03190-f003:**
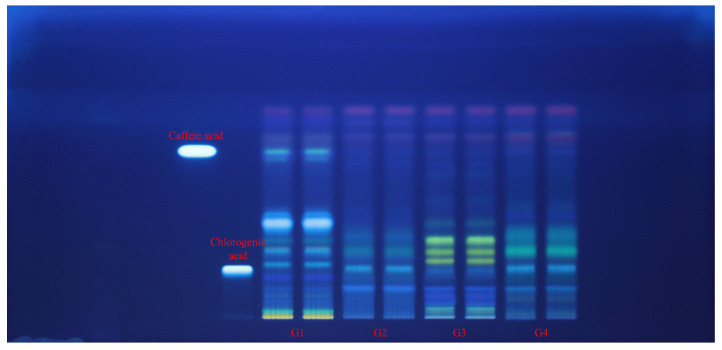
HPTLC fingerprint (365 nm, without derivatization) for *Geranium* (G1–G4) samples and phenolic acid reference bands.

**Figure 4 plants-14-03190-f004:**
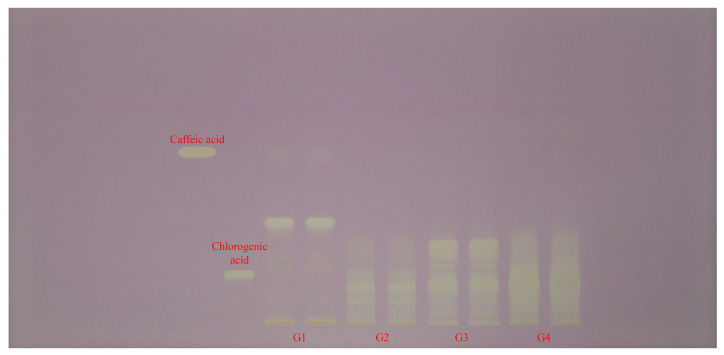
HPTLC–DPPH fingerprint (white light) for *Geranium* (G1–G4) samples and phenolic acid reference bands.

**Figure 5 plants-14-03190-f005:**
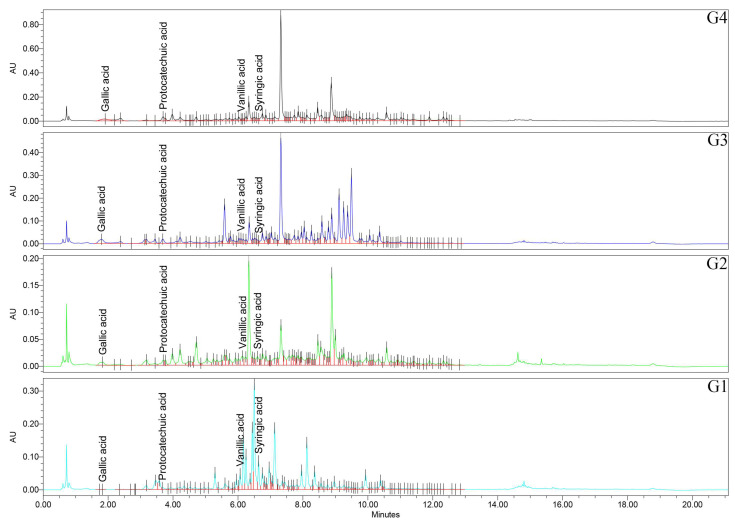
UHPLC/UV (265 nm) chromatograms for *Geranium* (G1–G4) samples. UHPLC: ultra-high-performance liquid chromatography; UV: ultraviolet.

**Figure 6 plants-14-03190-f006:**
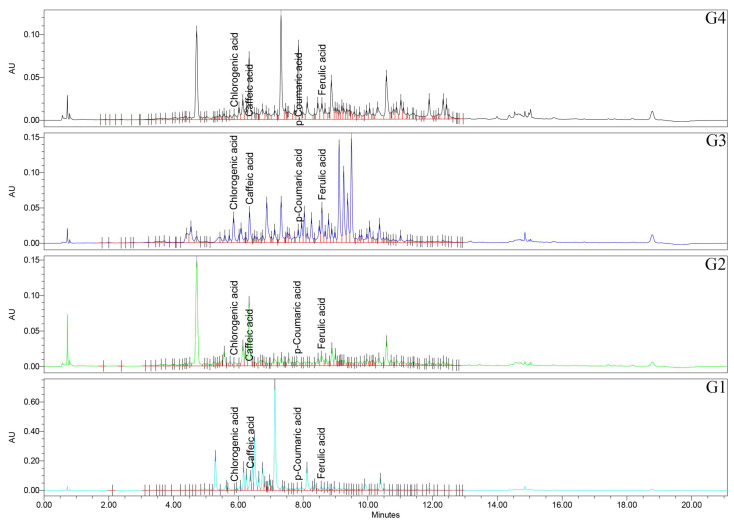
UHPLC/UV (325 nm) chromatograms for *Geranium* (G1–G4) samples.

**Table 1 plants-14-03190-t001:** Values (mean ± SD) of in vitro antioxidant (ABTS, DPPH, and FRAP), TPC and TFC assays for analyzed *Geranium* (G1–G4) samples.

Sample	ABTS IC_50_ (mg/mL)	DPPH IC_50_ (mg/mL)	FRAP (mM Fe^2+^)	TPC (mg GAE/g)	TFC (mg QE/g)
G1	1.283 ± 0.048 ^ns^	0.664 ± 0.029 ^ns^	27.727 ± 0.762 ****	68.170 ± 1.520 ****	5.462 ± 0.200 ^ns^
G2	0.640 ± 0.023 ^ns^	0.344 ± 0.014 ^ns^	38.049 ± 1.484 ****	82.930 ± 2.670 ****	4.060 ± 0.090 ^ns^
G3	0.404 ± 0.014 ^ns^	0.245 ± 0.015 ^ns^	52.483 ± 1.818 ****	120.360 ± 2.730 ****	6.716 ± 0.234 ^ns^
G4	0.321 ± 0.011 ^ns^	0.170 ± 0.006 ^ns^	60.492 ± 1.934 ****	140.140 ± 3.480 ****	5.967 ± 0.162 ^ns^

Statistical analysis was performed by two-way ANOVA followed by Tukey’s multiple comparison test: ns: not significant; **** *p* < 0.0001. ABTS: 2,2′-Azino-*bis*(3-ethylbenzothiazoline-6-sulfonic acid); ANOVA: analysis of variance; DPPH: 2,2-Diphenyl-1-picrylhydrazyl; FRAP: ferric-reducing antioxidant power; G1: *G. dissectum*; G2: *G. lucidum*; G3: *G. pussilum*; G4: *G. robertianum*; GAE: gallic acid equivalents; IC_50_: half-maximal inhibitory concentration; QE: quercetin equivalents; SD: standard deviation; TFC: total flavonoid content; TPC: total phenolic content.

**Table 2 plants-14-03190-t002:** Concentrations of phenolic acids quantified in *Geranium* (G1–G4) samples.

Sample	Caffeic Acid (μg/g)	Chlorogenic Acid (μg/g)	*p*-Coumaric Acid (μg/g)	Ferulic Acid (μg/g)	Gallic Acid (μg/g)	Protocatechuic Acid (μg/g)	Syringic Acid (μg/g)	Vanillic Acid (μg/g)
G1	650.928 ± 23.223 ****	87.821 ± 1.956 ^ns^	280.226 ± 5.703 **	112.034 ± 2.915 **	0.000 ± 0.000 ****	198.007 ± 6.636 ****	459.751 ± 16.764 ****	270.588 ± 9.994 ****
G2	518.406 ± 16.950 ****	62.373 ± 2.463 ^ns^	129.846 ± 4.912 ****	71.654 ± 1.787 ***	91.316 ± 2.243 ****	401.116 ± 8.381 ****	162.736 ± 3.340 ****	183.357 ± 5.725 ****
G3	256.377 ± 6.405 ****	283.606 ± 8.806 ****	244.255 ± 6.277 **	155.413 ± 5.988 ****	331.612 ± 8.327 ****	709.509 ± 19.062 ****	60.062 ± 1.454 ****	130.633 ± 3.801 ****
G4	411.969 ± 14.485 ****	87.959 ± 2.377 ^ns^	971.760 ± 25.246 ****	84.219 ± 3.186 ***	425.105 ± 11.872 ****	1236.165 ± 40.646 ****	95.272 ± 3.084 **	77.700 ± 2.101 ****

Values are mean ± SD (*n* = 3 independent extracts, each analyzed in triplicate). Statistical analysis was performed using two-way ANOVA followed by Tukey’s multiple comparisons test: ** *p* < 0.01; *** *p* < 0.001; **** *p* < 0.0001; ns: not significant. Multiple symbols in a cell indicate varying levels of significance across pairwise comparisons for that compound.

**Table 3 plants-14-03190-t003:** Sample coding of plant material (*Geranium* spp.).

Sample	Species/Vegetal Product	Date/Collection Site (Southwest Romania Flora; Geographic Coordinates)	Voucher Specimen
G1	*G. dissectum*/*herba*	20 April 2024/Stroeşti Commune, Vâlcea County (45°4′39.65″ N, 23°54′34.16″ E)	GER-DIS-2024-0420-2
G2	*G. lucidum*/*herba*	16 April 2024/Băile Herculane City, Caraş Severin County (44°53′54.69″ N, 22°25′46.91″ E)	GER-LUC-2024-0416-1
G3	*G. pussilum*/*herba*	28 April 2024/Cârcea Commune, Dolj County (44°16′29.84″ N, 23°52′32.89″ E)	GER-PUS-2024-0428-1
G4	*G. robertianum*/*herba*	16 April 2024/Băile Herculane City, Caraş Severin County (44°54′26.97″ N, 22°25′55.47″ E)	GER-ROB-2024-0416-2

## Data Availability

The original contributions presented in this study are included in the article. Further inquiries can be directed to the corresponding author.

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
