# Peer review of "Polyphenols Investigation and In Vitro Antioxidant Capacity of Romanian Wild-Grown Geranium spp. (Geraniaceae)"

_plants, 2025, doi:10.3390/plants14203190_

Round 1

Reviewer 1 Report

Comments and Suggestions for Authors

This manuscript investigates the phenolic composition and in vitro antioxidant activity of four wild-growing Geranium species from southwestern Romania. The authors analyzed total phenolic content (TPC), total flavonoid content (TFC), phenolic acid profiles using UHPLC/UV–MS, and antioxidant activities with DPPH, ABTS, and FRAP assays, complemented by HPTLC fingerprinting. The study is well-structured, the methodology is sound, and the results are clearly presented. It provides novel insights into lesser-studied Romanian Geranium species. Overall, the manuscript is scientifically valuable and suitable for publication after minor revisions.

  1. The introduction provides an extensive summary of the existing literature, but the novelty of the study is not sufficiently emphasized.It is recommended to highlight the significance of researching wild Geranium species from Romania. Please clarify the unique aspects of the study either at the end of the introduction or at the beginning of the discussion.
  2. The rationale for focusing on the four wild Geranium species from Romania is not explained.It would be helpful to provide justification for why these specific species were selected for the study.
  3. Provide exact collection locations (coordinates), precise collection dates, and the developmental stages of the plant material to ensure reproducibility.

In addition, clarify the extraction conditions, including solvent ratios, extraction time, and temperature parameters.

  1. The discussion of TFC and antioxidant capacity is incomplete, particularly regarding the dominance of non-flavonoid compounds.It is recommended to include a speculative mechanism or literature support to explain why non-flavonoid polyphenols dominate in FRAP/TPC assays.

Author Response

Dear Reviewer,

First of all, we would like to address you many thanks for your accurate observations and valuable comments. We used all these and improved the paper accordingly.

All changes in the revised manuscript were highlighted on a yellow background.

The following changes have been made to the Manuscript (ID: plants-3894889):

Reviewer #1 questions/comments

This manuscript investigates the phenolic composition and in vitro antioxidant activity of four wild-growing Geranium species from southwestern Romania. The authors analyzed total phenolic content (TPC), total flavonoid content (TFC), phenolic acid profiles using UHPLC/UV–MS, and antioxidant activities with DPPH, ABTS, and FRAP assays, complemented by HPTLC fingerprinting. The study is well-structured, the methodology is sound, and the results are clearly presented. It provides novel insights into lesser-studied Romanian Geranium species. Overall, the manuscript is scientifically valuable and suitable for publication after minor revisions.

Comments 1:

  1. The introduction provides an extensive summary of the existing literature, but the novelty of the study is not sufficiently emphasized. It is recommended to highlight the significance of researching wild Geranium species from Romania. Please clarify the unique aspects of the study either at the end of the introduction or at the beginning of the discussion.

Response 1:

Thank you very much for your insightful comment. To clarify the study’s novelty, a paragraph has been added at the end of “1. Introduction” section emphasizing that this is the first comparative evaluation of wild Geranium spp. from southwestern Romania. The new text highlights the unique regional scope and methodological integration of the work. (See page 3, lines 124–132).

Comments 2:

  1. The rationale for focusing on the four wild Geranium species from Romania is not explained. It would be helpful to provide justification for why these specific species were selected for the study.

Response 2:

Thank you for pointing this out. A sentence has been added before the study aim in the “1. Introduction” section to explain that the four selected wild-grown Geranium spp. are widespread in southwestern Romania, easily collected under the same environmental conditions, and underrepresented in previous phytochemical studies. (See page 3, lines 109–113).

Comments 3:

  1. Provide exact collection locations (coordinates), precise collection dates, and the developmental stages of the plant material to ensure reproducibility.

Response 3:

Thank you very much for your valuable suggestion. The exact collection locations (accurate geographical coordinates), precise collection dates, and the developmental stage of the plant material have been provided to ensure reproducibility. (See page 12, lines 394 & 402; Table 3).

Comments 4:

  1. In addition, clarify the extraction conditions, including solvent ratios, extraction time, and temperature parameters.

Response 4:

Thank you very much for your observation. The extraction conditions, including solvent ratios, extraction time, and temperature parameters, are already described in detail in “4.3. Extraction Procedure” subsection of the manuscript. (See page 12, lines 423–428; page 13, lines 429 & 430).

Comments 5:

  1. The discussion of TFC and antioxidant capacity is incomplete, particularly regarding the dominance of non-flavonoid compounds. It is recommended to include a speculative mechanism or literature support to explain why non-flavonoid polyphenols dominate in FRAP/TPC assays.

Response 5:

Thank you very much for your helpful suggestion. In the revised manuscript, the discussion of TFC and antioxidant capacity has been expanded in “3.1. Correlation Between TPC, TFC and Antioxidant Capacity” subsection. We now clarify that the lack of significant variation in TFC across species indicates that non-flavonoid phenolics are the main contributors to reducing capacity. (See “3.1. Correlation Between TPC, TFC and Antioxidant Capacity” subsection).

Comments 6:

Quality of English Language

(x) The English could be improved to more clearly express the research.

Response 6:

Thank you very much for your observation. Some grammar, stylistic or spelling errors have been corrected throughout the entire manuscript.

Authors very much appreciated the encouraging, critical, and constructive comments on this manuscript by the Reviewer. The comments have been very thorough and useful in improving the manuscript.

We would like to thank the Reviewer again for taking the time to review our manuscript.

We have also introduced other additions/modifications that we hope will improve the quality of the manuscript:

▪ Figures 1c and 1d have been modified accordingly.

▪ Funding information has been added accordingly.

▪ All abbreviations have been defined the first time they appear in the text.

▪ Some grammar, stylistic or spelling errors have been corrected.

Kind regards,

Ludovic Everard BEJENARU, PhD

Reviewer 2 Report

Comments and Suggestions for Authors

This study investigates four wild species of Geranium spp. (G. dissectum, G. lucidum, G. pusillum, and G. robertianum) in southwestern Romania. Using DPPH, ABTS, and FRAP antioxidant assays, along with phytochemical analyses including TPC and TFC. Quantitative analysis of eight phenolic acids was performed via UHPLC/UV. The study investigated total phenolic content, total flavonoid content, phenolic acid profiles, and in vitro antioxidant activity. These findings contribute to a deeper understanding of the antioxidant properties and chemical composition of Geranium species, providing theoretical support for their applications in medicine, health care, and related fields. Although only four varieties were slightly insufficient in the exploration of correlation, the logic of the study was clear and the results were scientific and accurate. 

The detailed comments are as follows:

  1. The statistical analysis method mentioned in the study uses two-factor ANOVA. Why? This study seems to only involve the variable of variety. The results of the statistical analysis are suggested to be labeled in Figure 1 or Table 1. How to set up biological repetition? It is recommended to supplement it in the material section?
  2. The results of the significance of differences in Table 2 are suggested to be supplemented.
  3. The abstract and in Line 208 mentioned the use of UHPLC/UV-MS to quantify the phenolic acids, but in fact, MS was not used for qualitative or quantitative analysis, but for standard comparison method, which is suggested to be modified.
  4. Some results described in the manuscript do not match those shown in the figures. It is suggested to be check and revised. For example,1) in Line192-193 mentioned ’G3 sample exhibited a bright, low‐Rf  fluorescent zone at the chlorogenic acid position’. However, the G2 and G4 samples showed brighter bands at the same position as chlorogenic acid in the Figure 3; 2) In Line200-202 mentioned ‘the number and intensity of bleaching zones were greatest in G4 and G3 samples, distributed across mid and upper Rf regions’, but in Figure 4, the bleaching zones of the G4 are located in the middle and low regions; 3) in Line205-206 mentioned ‘while G4 sample displayed multiple strong bleached bands not co‐migrating with caffeic or chlorogenic acids, indicating additional radical‐scavenging constituents in that extract’, but in Figure 4, a distinct band was found in the middle region of G1 sample, which did not overlap with caffeic acid and chlorogenic acid, indicating that there was some unknown component with strong radical scavenging function in G1, which was worth further identification.
  5. The FRAP/TPC gradient was mentioned several times, but the relevant results are not shown. It is suggested to supplement them in Figure 1 or Table 1.

Author Response

Dear Reviewer,

First of all, we would like to address you many thanks for your accurate observations and valuable comments. We used all these and improved the paper accordingly.

All changes in the revised manuscript were highlighted on a yellow background.

The following changes have been made to the Manuscript (ID: plants-3894889):

Reviewer #2 questions/comments

This study investigates four wild species of Geranium spp. (G. dissectum, G. lucidum, G. pusillum, and G. robertianum) in southwestern Romania. Using DPPH, ABTS, and FRAP antioxidant assays, along with phytochemical analyses including TPC and TFC. Quantitative analysis of eight phenolic acids was performed via UHPLC/UV. The study investigated total phenolic content, total flavonoid content, phenolic acid profiles, and in vitro antioxidant activity. These findings contribute to a deeper understanding of the antioxidant properties and chemical composition of Geranium species, providing theoretical support for their applications in medicine, health care, and related fields. Although only four varieties were slightly insufficient in the exploration of correlation, the logic of the study was clear and the results were scientific and accurate.

The detailed comments are as follows:

Comments 1:

  1. The statistical analysis method mentioned in the study uses two-factor ANOVA. Why? This study seems to only involve the variable of variety. The results of the statistical analysis are suggested to be labeled in Figure 1 or Table 1. How to set up biological repetition? It is recommended to supplement it in the material section?

Response 1:

Thank you very much for your valuable suggestion. We clarify that two-way ANOVA was employed because the design incorporated two independent factors: (i) species (G1–G4) and (ii) assay type (ABTS, DPPH, FRAP, TPC, TFC). This allowed us to examine both the main effects of species and assays, as well as possible interactions. For clarity, “4.9. Statistical Analysis” subsection has been revised to explicitly state this rationale. Post hoc pairwise comparisons were conducted using Tukey’s test within each assay, and significance levels have now been annotated in Table 1. Biological replication has also been clarified in the “4.9. Statistical Analysis” subsection (n = 3 independent extractions per species, each analyzed in triplicate). (See page 15, lines 524–531; Table 1).

Comments 2:

  1. The results of the significance of differences in Table 2 are suggested to be supplemented.

Response 2:

Thank you very much for pointing this out. In the revised manuscript, Table 2 has been updated to include statistical significance annotations derived from Tukey’s multiple comparisons test. Significance levels are now indicated using an asterisk system (* p < 0.05; ** p < 0.01; *** p < 0.001; **** p < 0.0001; ns: Not significant), consistent with the revised format of Table 2. This allows readers to clearly identify which differences among species are statistically meaningful for each quantified phenolic acid. (See Table 2).

Comments 3:

  1. The abstract and in Line 208 mentioned the use of UHPLC/UV-MS to quantify the phenolic acids, but in fact, MS was not used for qualitative or quantitative analysis, but for standard comparison method, which is suggested to be modified.

Response 3:

Thank you very much for your observation. The manuscript has been revised to clarify that mass spectrometry (MS) was used solely for confirmation of standards and not for quantitative purposes. The text has been modified accordingly in “4.8. UHPLC Analysis of Phenolic Acids” subsection, and throughout the manuscript where necessary. (See page 1, lines 30 & 31).

Comments 4:

  1. Some results described in the manuscript do not match those shown in the figures. It is suggested to be check and revised. For example, 1) in Line 192-193 mentioned ’G3 sample exhibited a bright, low‐Rf ‘fluorescent zone at the chlorogenic acid position’. However, the G2 and G4 samples showed brighter bands at the same position as chlorogenic acid in the Figure 3; 2) In Line 200-202 mentioned ‘the number and intensity of bleaching zones were greatest in G4 and G3 samples, distributed across mid and upper Rf regions’, but in Figure 4, the bleaching zones of the G4 are located in the middle and low regions; 3) in Line 205-206 mentioned ‘while G4 sample displayed multiple strong bleached bands not co‐migrating with caffeic or chlorogenic acids, indicating additional radical‐scavenging constituents in that extract’, but in Figure 4, a distinct band was found in the middle region of G1 sample, which did not overlap with caffeic acid and chlorogenic acid, indicating that there was some unknown component with strong radical scavenging function in G1, which was worth further identification.

Response 4:

Thank you very much for carefully noting the discrepancies between the textual descriptions and the HPTLC figures. In the revised manuscript, “2.4. HPTLC Fingerprinting and Effect-Directed DPPH Assay” subsection has been corrected to ensure full consistency with Figures 2–4. Specifically, the descriptions of the caffeic and chlorogenic acid bands have been revised, adjusted the comparative statements about bleaching intensities among species. (See page 6, lines 196–200, 203 & 204; page 7, lines 213 & 214).

Comments 5:

  1. The FRAP/TPC gradient was mentioned several times, but the relevant results are not shown. It is suggested to supplement them in Figure 1 or Table 1.

Response 5:

Thank you for your valuable feedback on FRAP/TPC gradient. The FRAP and TPC gradients (G4 > G3 > G2 > G1) are already visualized in the manuscript in Figure 1b (FRAP assay) and Figure 1c (TPC assay). (See Figure 1, b and c).

Comments 6:

Quality of English Language

(x) The English could be improved to more clearly express the research.

Response 6:

Thank you very much for your observation. Some grammar, stylistic or spelling errors have been corrected throughout the entire manuscript.

Authors very much appreciated the encouraging, critical, and constructive comments on this manuscript by the Reviewer. The comments have been very thorough and useful in improving the manuscript.

We would like to thank the Reviewer again for taking the time to review our manuscript.

We have also introduced other additions/modifications that we hope will improve the quality of the manuscript:

▪ Figures 1c and 1d have been modified accordingly.

▪ Funding information has been added accordingly.

▪ All abbreviations have been defined the first time they appear in the text.

▪ Some grammar, stylistic or spelling errors have been corrected.

Kind regards,

Ludovic Everard BEJENARU, PhD

Reviewer 3 Report

Comments and Suggestions for Authors

Comments for authors:

I would like to point out a small aspect regarding the term “antioxidant activity.” In my opinion, in the context of this manuscript, it would be more accurate to use the term “antioxidant capacity,” since the study refers to an extract containing multiple antioxidant compounds. The term “antioxidant activity” is usually used for a single isolated compound. Please correct this throughout the manuscript.

Lines 120-128 -I would like to point out that the opening paragraph of the Results section reads more like a concluding or discussion statement rather than an objective presentation of results. Expressions such as “coherent pattern,” “separate the species cleanly,” and “shows sharp, compound-specific differences” give the impression of interpretation rather than data description. I suggest rephrasing this paragraph to maintain a more neutral and descriptive tone appropriate for the Results section.

Lines 130-136 - The description of the ABTS and DPPH results appears rather interpretative and stylistically elaborate for the Results section. Phrases such as “virtually the same ranking,” “reinforcing the interpretation,” or “captured a common, narrow dynamic range” sound more appropriate for the Discussion. I recommend simplifying the wording and maintaining a more neutral, factual tone consistent with the presentation of experimental data.

The same data seem to be presented both in graphical form (Figure 1) and in tabular form (Table 1), which might be redundant. It would be sufficient to keep one of the two presentations or to ensure that each provides distinct information.

In addition, for the antioxidant assays (DPPH and ABTS/TEAC), the IC₅₀ units should be clearly specified. If Trolox was used as the reference standard, the results should be expressed as mg Trolox equivalents per mL (mg Trolox/mL) or otherwise indicate which standard was applied.

The statement, “FRAP assay cleanly differentiated every pair of species” seems too strong and not fully supported by the data. According to Table 1 and Figure 1, pairwise contrasts were reported as non-significant (p > 0.05). Therefore, statistical evidence does not confirm that all species were clearly differentiated based on FRAP results.

There seems to be some inconsistency regarding the statistical interpretation of the TPC and FRAP results. The authors state that “all six pairwise TPC contrasts were significant (p < 0.0001),” yet neither the figure nor the table indicate the statistical test performed or the corresponding significance markers. The description in the Figure 1 caption also does not support this claim. Therefore, the statement about all pairwise contrasts being significant should be verified or clarified.

The interpretation of the HPTLC results is mainly descriptive. The authors should specify the number of detected bands and the corresponding Rf values. Identification of phenolic acids based solely on visual co-migration is not sufficient; quantitative or semi-quantitative data (e.g., densitometry) would improve the reliability of this section.

Lines 209-212 - The description of LC-MS results is rather vague. Expressions such as “track the FRAP/TPC gradient” or “oppose that gradient” are unclear and should be replaced with specific quantitative relationships (e.g., positive or negative correlations). It would also be useful to indicate which phenolic acids belong to each group.

The statistical significance for the LC-MS results is mentioned only in the text, without being indicated in Table 2. Including statistical markers (e.g., different letters for p < 0.05) in the table would make the interpretation of significant differences much clearer and more transparent for readers.

The section entitled “Structure–Activity Correlation” seems overstated. The associations described between phenolic acids and antioxidant parameters are not supported by a clear statistical correlation analysis (e.g., Pearson or Spearman coefficients). The expressions “positive association” or “alignment with FRAP/TPC” are vague, and with only four species analyzed, such trends cannot be considered significant correlations. A more rigorous correlation analysis or a clarification of the statistical method would strengthen this section.

It would be useful to include the geographical location (coordinates) of the collection sites in Table 3 to enhance reproducibility and ecological context.

There is an inconsistency between the units reported in the Materials and Methods section (“µg GAE/mL extract”) and those in the results table (“mg GAE/g”). The units should be consistent throughout the manuscript, and the authors should clarify whether the values are expressed per volume of extract or per gram of dry sample. The same for TFC.

Author Response

Dear Reviewer,

First of all, we would like to address you many thanks for your accurate observations and valuable comments. We used all these and improved the paper accordingly.

All changes in the revised manuscript were highlighted on a yellow background.

The following changes have been made to the Manuscript (ID: plants-3894889):

Reviewer #3 questions/comments

Comments 1:

I would like to point out a small aspect regarding the term “antioxidant activity.” In my opinion, in the context of this manuscript, it would be more accurate to use the term “antioxidant capacity,” since the study refers to an extract containing multiple antioxidant compounds. The term “antioxidant activity” is usually used for a single isolated compound. Please correct this throughout the manuscript.

Response 1:

Thank you very much for your observation. The manuscript has been revised accordingly. (See page 1, lines 2 & 44; page 9, lines 254 & 276; page 13, line 438; page 14, line 489).

Comments 2:

Lines 120-128 - I would like to point out that the opening paragraph of the Results section reads more like a concluding or discussion statement rather than an objective presentation of results. Expressions such as “coherent pattern,” “separate the species cleanly,” and “shows sharp, compound-specific differences” give the impression of interpretation rather than data description. I suggest rephrasing this paragraph to maintain a more neutral and descriptive tone appropriate for the Results section.

Response 2:

Thank you very much for your helpful suggestion. In the revised manuscript, the beginning of “2. Results” section has been rephrased to adopt a more neutral and descriptive style, focusing strictly on the presentation of data without interpretation. This adjustment ensures that interpretative statements are reserved for “3. Discussion” section. (See page 3, lines 134–138).

Comments 3:

Lines 130-136 - The description of the ABTS and DPPH results appears rather interpretative and stylistically elaborate for the Results section. Phrases such as “virtually the same ranking,” “reinforcing the interpretation,” or “captured a common, narrow dynamic range” sound more appropriate for the Discussion. I recommend simplifying the wording and maintaining a more neutral, factual tone consistent with the presentation of experimental data.

Response 3:

Thank you for pointing this out. In the revised manuscript, “2.1. ABTS and DPPH IC50” subsection has been rephrased in a more neutral and descriptive manner. The updated text now reports the IC50 ranges for both assays, states that the ranking of species was similar, and specifies that pairwise comparisons did not reveal statistically significant differences, without interpretative language. (See page 3, lines 140–144).

Comments 4:

The same data seem to be presented both in graphical form (Figure 1) and in tabular form (Table 1), which might be redundant. It would be sufficient to keep one of the two presentations or to ensure that each provides distinct information.

Response 4:

Thank you very much for your suggestion. Both Figure 1 and Table 1 have been retained because they provide complementary information: Figure 1 illustrates the comparative trends among species, while Table 1 presents the exact values with statistical significance annotations. This ensures that readers can see both the overall patterns and the detailed statistical outcomes.

Comments 5:

In addition, for the antioxidant assays (DPPH and ABTS/TEAC), the IC₅₀ units should be clearly specified. If Trolox was used as the reference standard, the results should be expressed as mg Trolox equivalents per mL (mg Trolox/mL) or otherwise indicate which standard was applied.

Response 5:

Thank you very much for your observation. We would like to clarify that for IC50 determinations in ABTS and DPPH assays, no external standard is required. IC50 represents the concentration of the sample extract that produces 50% inhibition of the radical signal. It is derived directly from the dose–response curve of each extract, expressed in mg/mL, without reference to a calibration standard.

Comments 6:

The statement, “FRAP assay cleanly differentiated every pair of species” seems too strong and not fully supported by the data. According to Table 1 and Figure 1, pairwise contrasts were reported as non-significant (p > 0.05). Therefore, statistical evidence does not confirm that all species were clearly differentiated based on FRAP results.

Response 6:

Thank you for your insightful comment. We agree that the statement “FRAP assay cleanly differentiated every pair of species” was too strong. “2.2. FRAP Assay” subsection has been rephrased to present a more neutral description, stating only that FRAP values differed significantly among species. (See page 5, lines 162–167).

Comments 7:

There seems to be some inconsistency regarding the statistical interpretation of the TPC and FRAP results. The authors state that “all six pairwise TPC contrasts were significant (p < 0.0001),” yet neither the figure nor the table indicate the statistical test performed or the corresponding significance markers. The description in the Figure 1 caption also does not support this claim. Therefore, the statement about all pairwise contrasts being significant should be verified or clarified.

Response 7:

Thank you for your valuable feedback on the statistical interpretation of the TPC and FRAP results. In the revised manuscript, statistical significance has been annotated directly in Table 2 using the asterisk notation (* p < 0.05; ** p < 0.01; *** p < 0.001; **** p < 0.0001; ns: Not significant). This makes the pairwise differences in phenolic acid concentrations clear and directly visible to the reader. (See Table 2).

Comments 8:

The interpretation of the HPTLC results is mainly descriptive. The authors should specify the number of detected bands and the corresponding Rf values. Identification of phenolic acids based solely on visual co-migration is not sufficient; quantitative or semi-quantitative data (e.g., densitometry) would improve the reliability of this section.

Response 8:

Thank you for pointing this out. Detailed quantification of individual compounds by HPTLC was not within the scope of this study, as we employed UHPLC-based analysis for quantitative purposes. HPTLC was used here in a complementary role to visualize banding patterns and support the UHPLC/UV–MS results, rather than to provide quantitative data.

Comments 9:

Lines 209-212 - The description of LC-MS results is rather vague. Expressions such as “track the FRAP/TPC gradient” or “oppose that gradient” are unclear and should be replaced with specific quantitative relationships (e.g., positive or negative correlations). It would also be useful to indicate which phenolic acids belong to each group.

Response 9:

Thank you very much for your suggestion. “2.5. Phenolic Acids Profile (UHPLC/UV–MS Analysis)” subsection has been revised to replace vague expressions. (See page 7, lines 220–223).

Comments 10:

The statistical significance for the LC-MS results is mentioned only in the text, without being indicated in Table 2. Including statistical markers (e.g., different letters for p < 0.05) in the table would make the interpretation of significant differences much clearer and more transparent for readers.

Response 10:

Thank you very much for your observation. In the revised manuscript, the correlation annotations have been added directly into Table 2.

Comments 11:

The section entitled “Structure–Activity Correlation” seems overstated. The associations described between phenolic acids and antioxidant parameters are not supported by a clear statistical correlation analysis (e.g., Pearson or Spearman coefficients). The expressions “positive association” or “alignment with FRAP/TPC” are vague, and with only four species analyzed, such trends cannot be considered significant correlations. A more rigorous correlation analysis or a clarification of the statistical method would strengthen this section.

Response 11:

Thank you for your thoughtful suggestion regarding the structure–activity correlation. The title of “2.6. Structure–Activity Correlation” subsection has been changed to “2.6. Relationships Between Phenolic Profiles and Antioxidant Capacity” to better reflect the scope of our study. This subsection now emphasizes that the reported associations represent our findings for the Geranium spp. analyzed, rather than general mechanistic conclusions. (See “2.6. Relationships Between Phenolic Profiles and Antioxidant Capacity” subsection).

Comments 12:

It would be useful to the geographical location (coordinates) of the collection sites in Table 3 to enhance reproducibility and ecological context.

Response 12:

Thank you for pointing this out. The geographical locations (coordinates) of the collection sites have been included in Table 3 to enhance reproducibility and ecological context. (See Table 3).

Comments 13:

There is an inconsistency between the units reported in the Materials and Methods section (“µg GAE/mL extract”) and those in the results table (“mg GAE/g”). The units should be consistent throughout the manuscript, and the authors should clarify whether the values are expressed per volume of extract or per gram of dry sample. The same for TFC.

Response 13:

Thank you very much for your observation. The manuscript has been revised accordingly (See page 14, lines 475, 476 & 487.

Authors very much appreciated the encouraging, critical, and constructive comments on this manuscript by the Reviewer. The comments have been very thorough and useful in improving the manuscript.

We would like to thank the Reviewer again for taking the time to review our manuscript.

We have also introduced other additions/modifications that we hope will improve the quality of the manuscript:

▪ Figures 1c and 1d have been modified accordingly.

▪ Funding information has been added accordingly.

▪ All abbreviations have been defined the first time they appear in the text.

▪ Some grammar, stylistic or spelling errors have been corrected.

Kind regards,

Ludovic Everard BEJENARU, PhD

Round 2

Reviewer 3 Report

Comments and Suggestions for Authors

I would like to thank the authors for their thorough revision and thoughtful responses to the reviewers’ comments. Both clarity and scientific rigor have clearly improved the manuscript.